# Surface-induced water crystallisation driven by precursors formed in negative pressure regions

**Gang Sun** [1,2] **& Hajime Tanaka** [3,4] ✉

Ice nucleation is a crucial process in nature and industries; however, the role of the free surface of water in this process remains unclear. To address this, we investigate the microscopic freezing process using brute-force molecular dynamics simulations. We discover that the free surface assists ice nucleation through an unexpected mechanism. The surface-induced negative pressure enhances the formation of local structures with a ring topology characteristic of Ice 0-like symmetry, promoting ice nucleation despite the symmetry differing from ordinary ice crystals. Unlike substrate-induced nucleation via water-solid interactions that occurs directly on the surface, this negative-pressure-induced mechanism promotes ice nucleation slightly inward the surface. Our findings provide a molecular-level understanding of the mechanism and pathway behind free-surface-induced ice formation, resolving the longstanding debate. The implications of our discoveries are of substantial importance in areas such as cloud formation, food technology, and other fields where ice nucleation plays a pivotal role.

Water freezing phenomenon is ubiquitous in nature[1–4], science[5] and technology[6], such as atmospheric clouds[5], biological cells[7], meteors[8] and even air-conditioner applications. For instance, the radiative properties of clouds are determined by the formation of ice particles[9]. In the real world, most water exists in forms with interfaces, especially the free surface, such as water droplets in clouds and on all kinds of objects in daily life. Therefore, understanding the role of the free surface in ice nucleation and the microscopic ice nucleation mechanism at the molecular level is crucial from both fundamental and application perspectives. However, it is still challenging to directly detect the ice nucleation events at their inception since the necessary temporal and spatial resolution still needs to be improved in experiments. Tabazadeh et al.[10,11] proposed surface-induced crystallisation based on classical nucleation theory. Nucleation events near the free water surface are favoured thermodynamically than in bulk if $\sigma_{sv} - \sigma_{lv} > \sigma_{ls}$, with $\sigma_{sv}$, $\sigma_{lv}$, and $\sigma_{ls}$, the solid-liquid, solid-vapour, and liquid-vapour interface tensions, respectively. Due to the challenges of

measuring the interface tension, it is difficult to confirm whether the criterion works for water. Furthermore, in later years, more efforts on surface freezing provide results consistent and inconsistent with this theory[12,13], resulting in controversy until now. More seriously, whether we can use the macroscopic equilibrium properties to predict the nonequilibrium microscopic process, such as crystal nucleation, is unclear.

For addressing the problems of surface freezing in water, molecular dynamics (MD) simulations are powerful alternatives that make it possible to access microscopic information with a molecular-level resolution. However, the computational studies have also been inconclusive due to their dependence on the force fields, system sizes, and even ice molecule identification[14,15]. For instance, it was reported that for a free-standing water film with a six-site force field on the nanoscale, ice crystals likely nucleate near the surface[16–18] due to the lack of electrostatic neutrality. However, this result has not been reproduced in larger systems and is now speculated due to the finite-

[1]Social Cooperation Research Department "Frost Protection Science", Institute of Industrial Science, The University of Tokyo, 4-6-1 Komaba, Meguro-ku, Tokyo, Japan. [2]Center for Advanced Quantum Studies, Department of Physics, Beijing Normal University, Beijing, China. [3]Department of Fundamental Engineering, Institute of Industrial Science, The University of Tokyo, 4-6-1 Komaba, Meguro-ku, Tokyo, Japan. [4]Research Center for Advanced Science and Technology, The University of Tokyo, 4-6-1 Komaba, Meguro-ku, Tokyo, Japan. ✉e-mail: tanaka@iis.u-tokyo.ac.jp

system-size effects[15,19], although this needs to be confirmed. For all-atom models incorporating charges, such as TIP4P and TIP4P/ice[20,21], exploring the statistical aspects of ice nucleation through brute-force simulations is currently a formidable task. This is primarily due to the difficulty in acquiring trajectories that are both long enough and numerous enough to provide meaningful insights.

Concerning these difficulties, the coarse-grained monoatomic water model, mW water[22], significantly improves the computational efficiency due to the absence of long-range Coulomb electrostatic interactions. Besides, the mW water model can also achieve remarkable success in reproducing the features of water's energetics, thermodynamics, and structures[22,23]. For example, the mW model well captures the water's locally favoured tetrahedral arrangements and the relevant ice forms at ambient pressure[24], such as hexagonal Ice, cubic Ice, and Ice 0. Haji-Akbari et al.[14,15,19] and Li et al.[25] estimated the ice nucleation rates in free-standing films and nanodroplets of the mW water with a forward-flux sampling method. Both groups found that despite the system satisfying the criterion, $\sigma_{sv} - \sigma_{lv} > \sigma_{ls}$, surface freezing is unfavoured in the mW water, which is against Tanazadeh's theory[10,11]. Recently, Hayton et al.[26] also reached the same conclusion using a seeding technique. Specifically, they found that nucleation remains bulk-like in films thicker than ~ 3.5 nm, but the nucleation rate starts to decrease below a thickness of ~ 3.5 nm at 220 K.

Controversially, the conclusion drawn from the investigation of the realistic water model, TIP4P/ice[27], using forward-flux sampling[19], starkly contradicts the conclusions for the mW water model described above. This study reported a remarkable seven orders of magnitude enhancement in ice nucleation for 4-nm films compared to the bulk. However, perplexingly, their observations indicated that freezing initiates away from the air-water interface, introducing a counter-intuitive aspect to the results. On this issue, Hayton et al.[26] claimed that the apparent discrepancy between the mW and TIP4P/ice water models can be resolved through consistent treatment of truncated interactions between homogeneous and inhomogeneous systems when using a model with a long-range potential, such as TIP4P/ice.

Consequently, the current understanding remains far from conclusive. In this context, we highlight a potential challenge in applying seeding techniques and enhanced sampling algorithms to study crystal nucleation phenomena in a heterogeneous system.

Estimating the crystal nucleation rate through seeding techniques in a heterogeneous system can be tricky, as it is influenced by the selection of the seed's position and shape. On the other hand, techniques such as forward flux sampling and biased Monte Carlo heavily rely on the careful selection of a relevant order parameter to accurately describe the process of crystallisation.

In previous studies mentioned above, the local 6th-order bond orientational order parameter $q_6$ was employed for estimating the ice nucleation rate. However, a recent study on heterogeneous ice nucleation on a crystalline solid substrate, employing a forward flux sampling method, demonstrated[28] that the commonly adopted size-based method using the order parameter $q_6$, inadequately describes heterogeneous ice nucleation. Instead, the study highlighted the critical importance of considering the surface-induced geometric anisotropy of ice nuclei. Furthermore, it was reported that for other tetrahedral liquids like carbon[29] and silicon[30], the local $q_6$ is not a suitable order parameter for enhanced sampling techniques, with $q_3$ being more effective. These studies demonstrated that special care is necessary in selecting an appropriate order parameter when estimating the crystal nucleation rate, particularly when employing enhanced sampling techniques.

Given the unbiased simulation approach we employ to study ice nucleation, shape anisotropy may not pose a problem. However, a suitable order parameter remains crucial for accurately detecting crystal precursors and nuclei. Therefore, our aim is to identify an order parameter applicable to all tetrahedral liquids for this purpose. While

the order parameter $q_6$ can detect the six-fold symmetry of stable ices at ambient pressure, such as cubic and hexagonal ices, the ice formation pathway is not necessarily through a direct path but can involve Ice 0-like structures that contain five- or seven-membered rings[24], as manifested by the Ostwald step rule[31-33]. To detect these structures, we need a coarse-grained bond orientational order parameter with 12-fold symmetry, $Q_{12}$, capable of identifying all types of crystal structures in tetrahedral liquids. Indeed, we have confirmed that the local order parameter $q_6$ tends to overidentify ice molecules in the liquid phase, whereas the coarse-grained order parameter utilising the 12th-order bond orientational order parameter, $Q_{12}$, correctly identifies ice molecules and even outperforms $q_3$ (refer to Fig. 1a−c and Supplementary Fig. 1). This issue may not significantly affect the estimation of nucleation rates in bulk systems. However, in situations where surfaces induce ordering with symmetries other than 6-fold, as in our case, utilising $q_6$ for biasing could lead to incorrect conclusions.

To directly explore the microscopic kinetic pathway of ice nucleation, we conduct a series of extensive *brute-force* molecular dynamics (MD) simulations utilising the coarse-grained mW water model. To avoid potential curvature-induced effects, our initial focus is on studying free-standing water films featuring zero-mean-curvature vapour-liquid interfaces. Our approach involves a comprehensive analysis, employing topological ring analysis and the high-degree bond-orientational order parameter $Q_{12}$. The $Q_{12}$ order parameter can pick up any crystal forms of water, including Ice 0[24]. These tools allow us to quantitatively assess the relative arrangement of neighbouring water molecules, enabling the identification of ice-crystal precursors, colloquially referred to as preorders. These preorders exhibit a liquid state but possess crystal-like local orientational order. The utilisation of these advanced techniques helped us overcome the limitations associated with standard crystalline ice identification methods, ensuring an unbiased identification of preorders−precursors−from which ice crystals are anticipated to form. Our results furnish robust numerical evidence supporting free-surface-assisted ice nucleation, occurring near the free surface but slightly inward. Ice nucleation starts from the formation of Ice 0-like precursors near the free surface, but the precursors quickly transform more stable ordinary ices. Subsequently, we unravel an unconventional microscopic physical mechanism and kinetic pathway that facilitates ice nucleation in close proximity to the free surface through ice precursors. Finally, we demonstrate the relevance of our microscopic ice nucleation mechanism by extending our investigation to nanodroplets of the mW water model and thin films of the realistic TIP4P/ice water model[27].

## Results

### Preferential ice nucleation location in thin water films
First, we explain our strategy to attack a controversial and long-standing problem concerning whether a free water surface can facilitate ice nucleation or not. In earlier works[14,15,19,25], the ice nucleation rates of free-standing water films were estimated using enhanced sampling methods, such as forward-flux sampling[34] and umbrella sampling methods[35]. The cumulative transition probabilities were used to quantify the nucleation rates in both thin films and bulk water. However, the properties of the free surface are pretty complicated; for example, the surface structure is strongly distorted due to the symmetry breaking at the interface[14,15,19,25,36,37]. Accordingly, the dynamics near the free surface is also different from the bulk[36,38,39]. Hence, it has been considered reasonable if the ice nucleation rate is slower for a free-standing film than the bulk.

Instead, we ask whether a free water surface can induce nucleation and whether a preferential location of ice nucleation exists. To answer these questions, we directly detect the birth location of ice nuclei during the nucleation process by brute-force MD simulations (see Supplementary Movie 1). Figure 1d depicts the relative position where ice nuclei are born. It shows that the ice nuclei are still small initially

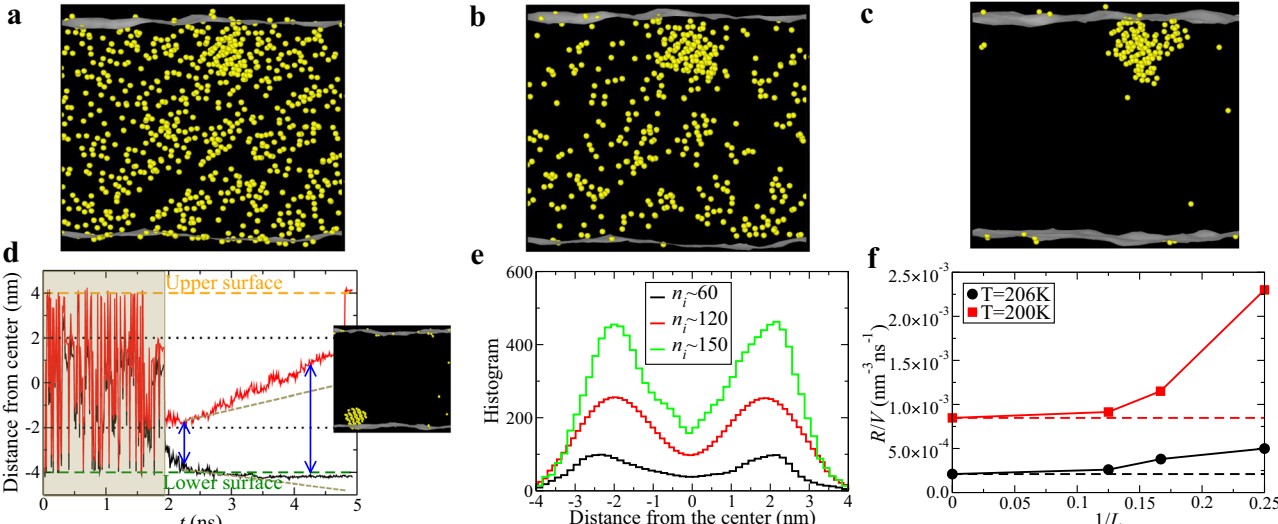

**Fig. 1 | Ice nucleation in a thin water film. a-c** Ice molecules observed in a thin water free-standing film with a thickness of $L$ = 8 nm, equilibrated at $T$=206 K, at $t$ = 2.63 ns, characterised by local orientational order parameter $q_6$ (**a**), local $q_3$ (**b**), and coarse-grained bond orientational order parameter $Q_{12}$ (**c**). Crystal molecules are identified based on the number of connections detected by the scalar product of the respective order parameter in each case. **d** A typical example of ice nucleation and growth in molecular dynamics simulations of a thin water film at $T$ = 206 K. The upper orange and lower green dashed lines indicate the geometric boundaries of the film. The regions between these lines and the dotted lines indicate the sub-surface regions. Red and black lines mark the edges of the largest crystalline ice cluster. Brown shading indicates the initial stage where small ice nuclei fluctuate throughout the water film. Inset: Snapshot of an ice nucleus in the thin water film during nucleation. The yellow particles indicate the ice molecules identified by the bond orientational order parameter $Q_{12}$. The free water surfaces are identified by the outmost molecules of the thin film (see the white surface meshes). **e** The distribution of ice clusters with various sizes along the thickness direction, which is obtained by averaging over 100 independent trajectories. **f** A comparison of the nucleation rate per volume, $R/V$, between bulk water and water films of thickness $L$ = 4, 6 and 8 nm. The statistical uncertainty associated with the estimated rate constant over 100 independent trajectories is quite small and does not affect the conclusions drawn from our investigation into the nucleation behaviour of water films (refer to Supplementary Fig. S2c).

and fluctuate across the water film. As the largest nucleus grows and reaches the critical nucleus size, the first ice crystal is born in the sub-surface region (see the image in Fig. 1d) and then increases its size linearly with time. We also show the microscopic dynamic process of ice nucleation in the water film in Supplementary Movie 1. It is suggestive of free-surface-assisted ice nucleation. We also show the histogram of ice crystals of different sizes as a function of the distance from the film centre in Fig. 1e. We observe that ice crystals are more likely to nucleate in the sub-surface region (not directly on the surfaces) than in the middle region, based on statistics from over 100 independent trajectories. The symmetric double-peak distribution of the ice nucleation probability (predominantly cubic ices) near the two surfaces of the water film strongly supports free-surface-induced ice nucleation.

Moreover, we directly measure the nucleation rates of both water films and bulk water. Nucleation is inherently a stochastic process; therefore, even for the same system, we must wait for varying time intervals to observe nucleation events. In this context, we determine the birth time of the first crystal, referred to as the induction time $t_{ind}$, for each trajectory by monitoring the time evolution of potential energy (refer to Supplementary Fig. 2a). Utilising the distribution of induction times $t_{ind}$, we estimate the probability for a single simulation trajectory to remain in the liquid state at time $t$, denoted as $P_{liq}(t)$. The results are shown in Supplementary Fig. 2b (see "Methods" for the definition). We quantify the nucleation rate $R$ by fitting a stretched exponential function to $P_{liq}(t)$ (see "Methods" for the details).

In Fig. 1f, we illustrate the comparison of nucleation rates per volume, $R/V$ (nm$^{-3}$s$^{-1}$), between bulk water and water films. Our calculated nucleation rate for bulk water aligns well with the results reported in refs. 25,38. Notably, the nucleation rate in the water film surpasses that of the bulk, exhibiting an increasing trend as the water film thickness decreases. This observation aligns with our visual evidence that ice crystals tend to form predominantly in the sub-surface

regions of a thin water film, as depicted in Fig. 1d and e. Here, it is worth noting that in thin films, crystal nucleation is influenced by dynamic capillary waves, a factor absent in quiescent bulk states. We plan to investigate the impact of this effect on crystal nucleation and growth in future studies.

Our findings diverge qualitatively from earlier results obtained using the same mW water model. In 2003, Lü et al.[38] demonstrated that the total nucleation rate of water films increases and approaches the bulk one when the thickness of films increases. They reported that the ice nucleation rate decreases by only about a factor of two for the thinnest film (4.72 nm) compared to the bulk. On the contrary, Haji-Akbari et al. reported a significantly more pronounced decrease in the nucleation rate by two to three orders of magnitude in 5-nm-thick films and a decline of seven orders of magnitude for 2.5-nm-thick films[14]. It is important to note that Lü et al.[38] and Haji-Akbari et al.[14] employed different enhanced sampling methods, namely, the mean first-passage time vs. forward-flux sampling, and different order parameters to identify ice molecules, i.e., the CHILL algorithm[40] vs. $q_6$[14,15,19], respectively. Additionally, a recent study by Hayton et al.[26], utilising a seeding technique, further supported the notion that the nucleation rate is lower for thin films compared to the bulk.

Now, we consider the reasons for the disparity in conclusions between our study and those mentioned above. As discussed earlier, sampling methods are highly sensitive to the choice of order parameters associated with crystals. The previous studies mentioned above utilised the CHILL algorithm or $q_6$ order parameters, assuming the relevant ice type to be ice I, i.e., cubic or hexagonal ices. However, as we will demonstrate later, this assumption might not be valid. For tetrahedral liquids, such as water, the most relevant order parameter is $Q_{12}$[24], involving its 16 neighbours, as it can detect any crystals favoured by local tetrahedral symmetry, including Ice 0 and Clathrate. We employ this order parameter to identify ice-like preorders and ice crystals.

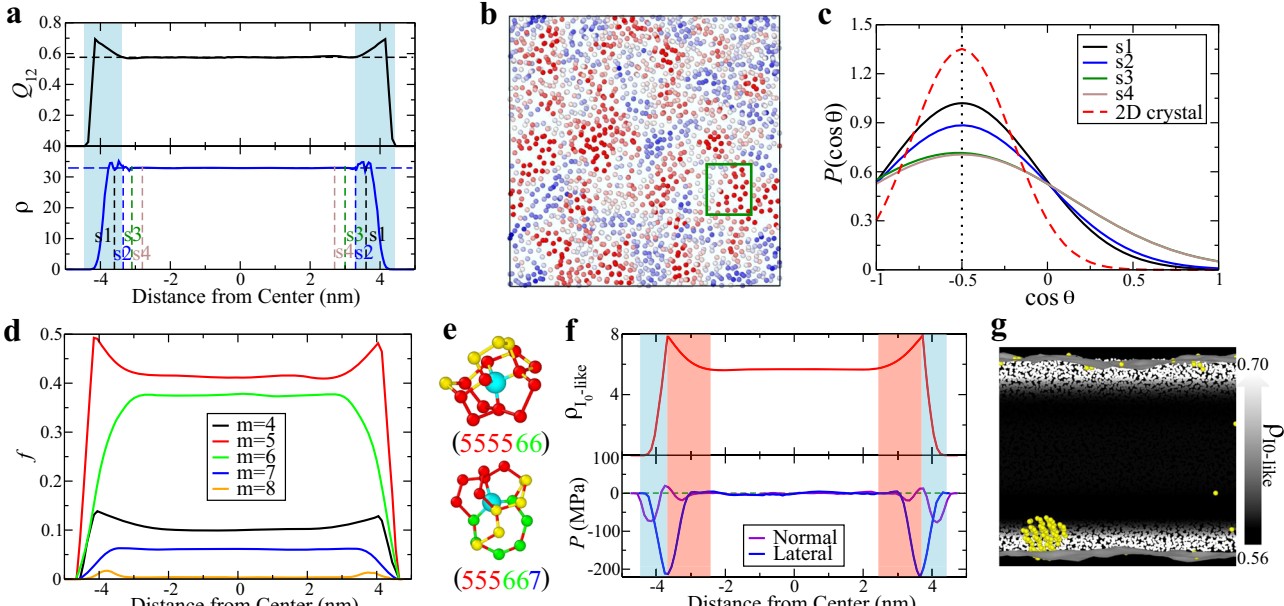

**Fig. 2 | Roles of Ice 0-like precursors in surface-induced ice nucleation. a** Spatial profiles of the bond orientational order parameter $Q_{12}$ and density $\rho$ across the water film at $T = 206$ K. The cyan shading indicates the outmost water surface layers. Four selected slabs from the outermost layer to the interior are labelled as s1, s2, s3, and s4. **b** Structure of the outermost layer, coloured by the 2D orientational order parameter $\bar{\lambda}_1$. The colour from blue to red indicates the low to the high value of $\bar{\lambda}_1$. **c** Distribution of $\cos\theta$ of water molecules in different slabs, s1, s2, s3 and s4, along with the one for 2D crystal. $\theta$ denotes the angle of a molecule with its two neighbours in the 2D plane parallel to the water surface. **d** Fraction $f$ of water molecules involved in $m$-membered rings across the direction normal to the water surface ($t = 1$ ns). **e** Ice 0 ring structures of local elementary building blocks, (5,5,5,5,6,6) and (5,5,5,7,6,6). **f** Number density $\rho_{I_0-like}$ of Ice 0-like water molecules identified by the ring analysis at $T = 206$ K. The pink-shaded region indicates the sub-surface region with rich Ice 0-like molecules. The pressure in both normal and lateral directions of water films is also presented. **g** Snapshot of ice nuclei with the critical nucleus size at $T = 206$ K. The ice molecules (yellow) are identified by $Q_{12}$. The density of Ice 0-like precursors $\rho_{I_0-like}$ is represented using a white background when the critical nucleus is born. The colour changes from black to white with an increase in the density of Ice 0-like precursors (see the colour bar).

Moreover, it is crucial to note that while seeding techniques may prove valuable for examining crystal nucleation behaviours, their efficacy in investigating the influence of the free surface on ice nucleation could be questionable. Ice nucleation near a free surface may follow a distinct kinetic pathway involving Ice 0-like precursors, adding complexity to the process. Unlike in bulk, the selection of the seed's position and shape may also play a crucial role in estimating the nucleation rate in the presence of symmetry-breaking surface fields.

Taking these factors into consideration, we assert that a direct approach—employing brute-force simulations—provides a more thorough and convincing estimation of the crystal nucleation rate in free-standing films, compared to enhanced sampling algorithms or seeding methods, even though this method demands more computation time.

**Surface-induced nucleation mechanism**

Due to its unique structural and mechanical properties, the water surface plays a crucial role in ice nucleation. For instance, Odendah et al.[41] recently reported that water and ice share similar characteristic structural features, implying the presence of nanometre-scale ice-like domains at the air-water interface. In our work, we also observe a layering phenomenon and ordered structure near the free water surface in Fig. 2a, suggesting a structured liquid at the surface. Upon closer examination of the first few layers of the water surface, we identify an ordered local structure (red region) in the 2D plane of the well-defined outermost two molecular layers, composed of five-, six-, and seven-membered rings (see the region inside the green square in Fig. 2b). A recent experimental study of 2D ice crystals formed on a solid substrate using scanning tunnelling microscopy (STM) indicates that the preordered structure of 2D hexagonal ice consists of abundant five-, six-, and seven-membered rings[42]. This implies that the water surface possesses a 2D hexagonal ice-like local structure. Moreover, by comparing the angular distribution of a water molecule with its neighbours in the 2D plane parallel to the water surface, we also observe pronounced peaks in the angular distribution of water molecules in the first two layers, s1 and s2 (see the cyan-shaded region in Fig. 2a), similar to the 2D hexagonal ice. The peak position in the three-body-angle distribution is at 120° (see Fig. 2c). Therefore, both previous results and our observations seem to indicate that the ice-like ordered water surface may aid in facilitating crystallisation, contrary to the notion of 'bad' ordering suppressing nucleation reported in previous works[7,14,15,19].

Consequently, assuming the presence of surface-induced precursors on the water surface appears to be physically reasonable. We use the 2D orientational order parameter $\bar{\lambda}_1$ (see "Methods" for the definition) to quantify the surface structural order and its correlation with ice nucleation in the sub-surface region seen in Fig. 1. As shown in Supplementary Fig. 3a and b, we also find highly ordered surface molecules (indicated by green beads), which strongly correlate with the ice precursors and ice clusters formed in the interior yet close to the surface. This result is consistent with the scenario of surface-induced ice nucleation in free-standing films. However, it is essential to note that ice nuclei are formed predominantly in the sub-surface regions, not directly on the free surfaces, as depicted in Fig. 1d and e. Hence, the critical question arises: whether the 2D ice-like order of the surfaces is responsible for ice nucleation, or if some other source plays a more dominant role in inducing ice nucleation in the sub-surface regions. Below, we will show that the latter is the case.

To address this question, analysing water structures in the sub-surface regions is crucial. In this context, we opt for ring analysis as an alternative to the conventional bond-orientational order parameter. This choice is motivated by the surface's potential to distort the symmetry of locally ordered structures in its vicinity, a phenomenon

absent in the bulk. A striking revelation emerges from our analysis: 5-membered rings are significantly more abundant at the surface than in the bulk and extend deep into the interior of the water film, reaching a characteristic length of 1.5 nm, as illustrated in Fig. 2d. The 5-membered ring exhibits a preference in energy over the 4-membered ring and a greater favorability in entropy than the 6-membered ring. This competition, coupled with the surface constraint, serves as the origin for the enrichment of 5-membered rings at the surface.

Russo et al.[24] reported the spontaneous creation of structural orders associated with a metastable ice form known as Ice 0, containing 5-, 6-, and 7-membered rings, in a supercooled state. These structures act as ice precursors by lowering the free-energy barrier of ice nucleation. Ice 0 structures exhibit two distinct local structural motifs[24]: (5,5,5,5,6,6) and (5,5,5,7,6,6), as depicted in Fig. 2e, where the numbers 5, 6, and 7 indicate the count of water molecules in rings including the central molecule as a member. It was shown[24] that Ice 0-like structures (see Fig. 2e) play a crucial role in the ice nucleation of bulk water. These structures are more readily formed in supercooled water than Ice I-type orders (cubic and hexagonal arrangements) since their ring distribution closely resembles that of liquid water. A molecule is classified as Ice 0-like if at least five of its six rings closely match the Ice 0 ring structures illustrated in Fig. 2e. These Ice 0-like preorders play a pivotal role in fostering the nucleation of ice crystals by substantially diminishing the crystal-liquid interfacial energy cost[24]. Ice 0-like structures are not stable, and as the order develops, they undergo rapid transformation into ordinary ices, predominantly in cubic ices, without forming Ice 0[24].

Interestingly, we observe an abundance of Ice 0-like preordered structures in the sub-surface regions. In Fig. 2f, the density of Ice 0-like precursors, $\rho_{I_0-like}$, in the pink-shaded region exhibits a prominent maximum near the position adjacent to the 2D ice-like surface layers (see the blue-shaded region). Notably, the characteristic thickness of this sub-surface region is ~1.5 nm, as observed in Fig. 2f. The pressure within the water film, both in the normal and lateral directions, is negative in these sub-surface regions (the pink-shaded regions). It is known[24] that for the bulk case, the fraction of Ice 0-like molecules increases as pressure decreases (see Supplementary Fig. 4). In connection with this, it is worth noting that the homogeneous ice nucleation rate is enhanced under negative pressures[43]. Hence, we deduce that the abundant Ice 0-like environments stem from the surface-induced negative pressure in the sub-surface regions. Ice nuclei emerge in these preordered liquid environments, specifically in highly concentrated Ice 0-like environments, as illustrated in Fig. 2g. This surface-induced preordering in liquid water provides a free-energy-based rationale for surface-induced ice nucleation. Interestingly, a similar phenomenon of surface-induced nucleation in a negative pressure region was reported for silicon[30], suggesting a general mechanism of surface-induced nucleation for tetrahedral liquids showing a density decrease upon solidification.

The above finding indicates that the primary role in surface-induced ice nucleation is played by Ice 0-like preordering in the sub-surface regions (the pink-shaded region in Fig. 2f) rather than 2D ice-like order on the surfaces (the blue-shaded region in Fig. 2f). Symmetry breaking at the surfaces not only induces layering, i.e., short-range translational order, leading to 2D ice-like order, but also stretches the surfaces, resulting in negative pressure in the sub-surface regions. While the former contributes to the induction of 2D ice-like order, the latter is more influential in enhancing Ice 0-like preorders. Our study suggests that the latter mechanism plays a more substantial role than the former in the formation of ice nuclei in water films. This observation may be attributed to the fact that, unlike the case of a solid substrate, the 2D ice order at the surface is not as robust due to surface fluctuations, i.e., capillary waves, which strongly perturb it.

Besides thermodynamic factors, dynamics is another crucial factor controlling crystal nucleation. The dynamics is also heterogeneously distributed in water films[38,44]. Specifically, the surface mobility of the mW water is slower than that of the interior (see Supplementary Fig. 5). For bulk water, Martin et al.[45] proposed that ice nucleation in supercooled water occurs in less mobile liquid regions. This phenomenon can be explained through two steps. Firstly, pre-ordered regions exhibit slower dynamics in a supercooled liquid[46], and secondly, preorders facilitate crystal nucleation because of the angular symmetry matching[33] that lowers the energy cost of the crystal-liquid interface[47]. The slower diffusion near the water surface suggested ordered surface structures, as shown in Supplementary Fig. 5b, providing additional support for the surface-induced ice nucleation mechanism described above.

## Ice nucleation in a water nanodroplet

A water droplet is one of the most common forms of liquid water in nature and is crucial for atmospheric science, such as clouds[5] and rains[8]. It is also of technological importance. A water droplet, characterised by a large surface-to-volume ratio, is widely employed to investigate the significance of the surface on water's properties, including its impact on dynamic behaviours and ice nucleation.

To further confirm the effects of the free surface on nucleation, we directly measure the spatial distribution of Ice 0-like and ice molecules and track their evolutions in water droplets. Remarkably, nucleation events in water droplets are not random; instead, ice nuclei exhibit a higher probability of occurring in positions close to the surface in nanodroplets, as illustrated in Fig. 3a (see also Supplementary Movie 2). This behaviour is evident when projecting the positions of molecules belonging to ice precursors and crystals onto a 2D cross-section map, as shown in Fig. 3b. The distribution of ice molecules along the radial direction, depicted in Fig. 3c, aligns with the 2D spatial map of ice molecules (see the inset of Fig. 3c). We also provide a microscopic view of the dynamic ice nucleation process in the water droplet at 150 K in Supplementary Movie 2. This result strongly supports the surface-assisted nucleation mechanism for water droplets, a topic that has been debated for decades. Based on Fig. 3a and b, we can conclude that ice nucleation in water droplets occurs heterogeneously in space, facilitated by the curved surface. Our structural analysis further reveals the formation of Ice 0-like precursors near the surface, as evidenced by the distributions of ring structures and Ice 0-like structures along the droplet's radial direction, as shown in Fig. 3d and e, respectively.

In water droplets, Ice 0-like structures are abundant in the sub-surface region and gradually decrease into the interior. Similar to water films, we observe a negative tangential pressure $P_T$ near the water droplet surface due to surface tension (see Fig. 3e), as reported by Li et al.[25]. We conclude that this surface-induced negative pressure is the physical reason for the rich Ice 0-like environments close to the droplet surface. As well-known, the surface curvature of a water droplet creates extra pressure called the Laplace pressure, $P_L = P_N - P_T$, which increases the pressure inside the droplet. As the droplet size decreases, the internal pressure increases ($P_L$), which tends to suppress the probability of a nucleation event in the droplet, as shown in Li et al.'s work[25]. However, due to the relatively low tangential pressure ($P_T$) close to the surface, ice nucleation still prefers to occur near the surface for the droplet size studied here.

## Comparison with TIP4P/Ice water model

As described above, the use of a coarse-grained water model (mW water in our case) provides us with the capability to directly observe the microscopic dynamics of ice nucleation, facilitated by its computational efficiency. However, there might be concerns regarding the reliability of this model in faithfully representing water properties, given that simulation outcomes can be influenced by the choice of force field and system size.

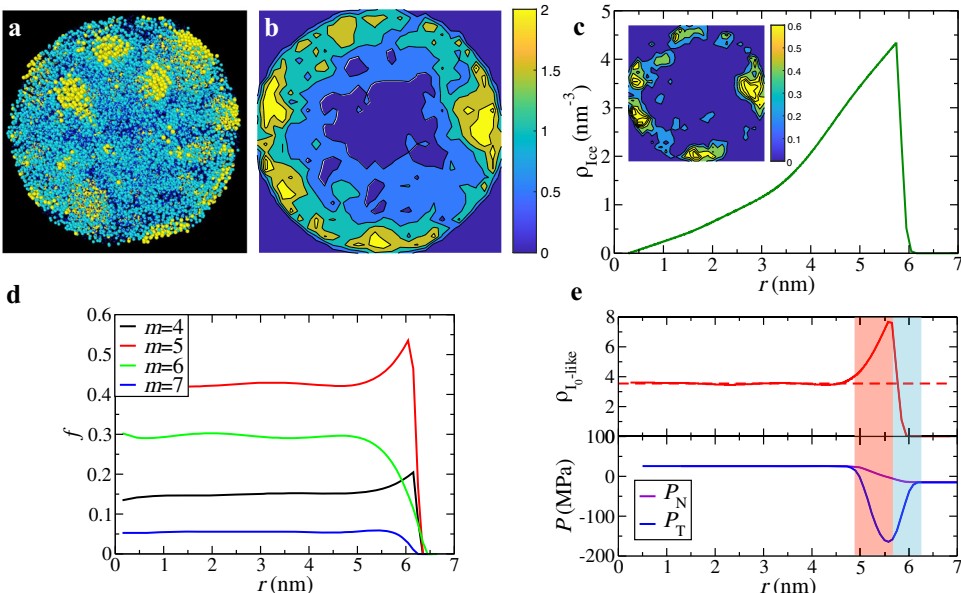

**Fig. 3 | Ice nucleation in a water nanodroplet at $T = 180$ K. a** Snapshot of ice molecules in a water droplet with a radius of 6 nm at $t = 21$ ns. The yellow particles indicate ice molecules, and the cyan particles indicate ice precursors. See also Supplementary Movie 2. **b** The distributions of ice preorders and ice molecules by angular averaging around the polar axis ($t = 21$ ns). **c** The number density $\rho_{\text{Ice}}$ of ice molecules along the radial direction of the water droplet at $t = 21$ ns as a function of the distance from the droplet centre $r$. Inset: The distribution of the ice molecules by angular averaging around the polar axis. **d** The dependence of the fraction $f$ of $m$-membered ring structures on the distance from the droplet centre $r$ at $t = 1$ ns. **e** The

spatial distribution of the density of Ice 0-like precursors ($\rho_{I_0 - like}$) and the normal ($P_N$) and tangential pressure ($P_T$) as a function of the distance from the droplet centre at $t = 1$ ns. The cyan-shaded region delineates the outermost water layers of the droplet, while the pink-shaded area denotes the sub-surface region abundant in Ice 0-like molecules. The pressure in both normal and lateral directions of water films is also presented. We can see that ice crystal formation (panel **c**) follows the distribution of the precursors (panel **e**), which results from the negative tangential pressure $P_T$ close to the surface of the water droplet.

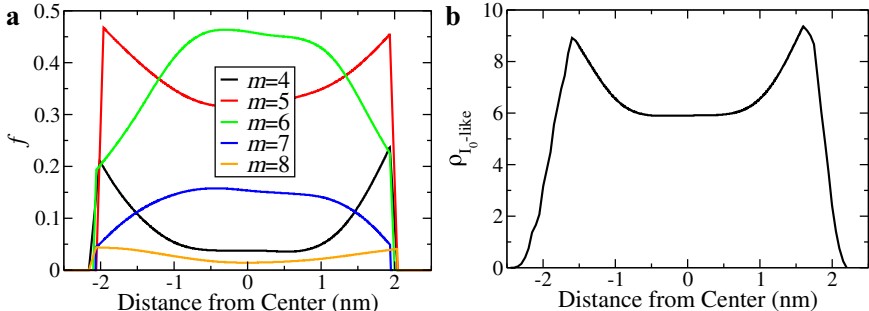

**Fig. 4 | Surface-induced structural ordering in the TIP4P/ice water film of 4-nm thickness at $T = 230$ K.** The profiles are analysed at $t = 1.0$ ns before ice nucleation occurs. **a** The fraction $f$ of molecules involved in $m$-membered rings along the

thickness direction. **b** The number density profile $\rho_{I_0 - like}$ of Ice 0-like water molecules along the thickness direction.

In this regard, the TIP4P/ice model stands out as one of the most accurate classical molecular models of water, particularly in describing ice formation. Nevertheless, the direct simulation of ice nucleation events using this water model proves challenging due to its computationally demanding nature. Recently, it has been demonstrated that crystal nucleation is initiated by the spontaneous formation of crystal precursors (or preordering) within a liquid state[33,46]. Consequently, our specific focus is directed towards exploring preordering rather than directly simulating nucleation events. To identify preorders, we employ ring analysis to capture the water structures near free surfaces, with the expectation that these structures would exhibit an Ice 0-like configuration, as observed in the results obtained with the mW water model. The ring and density distributions of Ice 0-like preordered structures are illustrated in Fig. 4a and b, respectively. Consistent with the mW water results (see Fig. 2), the fraction of the five-membered rings increases towards the surfaces, and the concentration of Ice 0-like

structures, formed by hydrogen bonds, exhibits distinct maxima in the sub-surface regions.

It is reasonable to anticipate that ice crystals would more readily be nucleated in regions with a high degree of Ice 0-like preorder, as observed in the mW water case presented in this study and the bulk case[24]. Thus, our TIP4P/ice model results provide robust support for surface-induced freezing in real water. The notion of 'bad' interfacial ordering in prior works[15] might be a misguided conclusion stemming from the usage of stable bulk ice orders (Ice c and h) as the critical structural order parameters to describe ice nucleation. These structures, associated with stable ices, differ significantly from those of liquid water (whether in the bulk or near the surface). Due to the high interfacial energy cost arising from the substantial structural differences between the crystal and liquid phases, direct nucleation of ordinary ices is not favoured. Instead, Ice 0-like precursors, exhibiting a ring distribution similar to that of liquid water, are demonstrated to be preferentially formed near the free water surfaces, acting as triggers

for ice nucleation. This two-step-like ordering pathway aligns with Ostwald's step rule of phase ordering[24].

## Discussion

Our work has unveiled a physical mechanism for free-surface-induced ice nucleation in water films and nanodroplets. In contrast to previous approaches that estimated cumulative transition probabilities based on the largest crystallite, we directly observed the microscopic kinetic pathway of ice nucleation using brute-force molecular dynamics simulations with the mW water model. Our findings indicate that ice nucleation events tend to occur near the surface region rather than in the middle, suggesting a higher probability of ice nucleation in thin films compared to bulk water.

While this observation aligns with previous computational studies of ice nucleation in TIP4P/ice model[15,19] that demonstrated a significant increase in the nucleation rate for free-standing films, the fundamental explanations differ. Previous studies argued that the confinement effect enhances homogeneous ice nucleation in the middle of a thin water film, characterising the order at the free surface as a 'bad' order and the order in the interior region as a 'good' order. This assignment appears self-paradoxical, given the high nucleation rate for a thin film compared to the bulk. Our study may provide a resolution to this issue.

We have revealed that the surface-tension-induced negative pressure contributes to the enrichment of the 5-membered ring environment in the sub-surface regions, promoting local structural ordering in the form of Ice 0-like precursors. These precursors exhibit unique ring topologies that include 5-, 6-, and 7-membered rings. Unlike conventional ice forms that consist solely of six-membered rings, such as $I_h$ and $I_c$, Ice 0-like structures exhibit significantly higher compatibility with liquid water compared to ordinary ice due to their topological resemblance, thereby reducing the interfacial energy required for ice nucleation. Additionally, they demonstrate an affinity for $I_c$, serving as precursors for ordinary ice[24]. Moreover, Ice 0-like structures, distinct from Ice 0 itself and lacking translational order, are hypothesised to act as precursors even in temperature ranges where Ice 0 is unstable. Local symmetry matching and the resulting transformability to a crystal are merely prerequisites for these structures to serve as precursors[33,46]. However, given that previous studies employing mW water have explored such temperature ranges[14,15,19,25,26], this aspect warrants further careful investigation.

Notably, the abundance of Ice 0-like precursors in the sub-surface regions, just inward from the free surfaces where negative pressure prevails, aligns with the locations exhibiting the highest ice nucleation probability. It is crucial to emphasise that this negative-pressure-induced mechanism fundamentally differs from substrate-induced heterogeneous nucleation, where ice nucleation occurs directly on the surface through interactions with the substrate.

The ordering of water induced by the free surface should be regarded as a 'good' order, as it actively promotes ice nucleation in close proximity to the free surfaces of water. We advocate for characterising ice nucleation in nature as 'heterogeneous (surface-induced)' rather than 'homogeneous'. These findings not only contribute to a fundamental comprehension of ice nucleation in water films and droplets but also hold significance for natural sciences such as climate science and various technological applications.

Here, we briefly explore the possibility of experimentally detecting surface-induced ice nucleation. In their study, Duft and Leisner compared the freezing probabilities of droplets with radii of 49 $\mu$m and 19 $\mu$m[12]. They concluded that homogeneous freezing is a volume-proportional process and that surface nucleation might only be significant, if at all, for much smaller droplets. As they acknowledged, in relatively large droplets, the volume of the thin surface region could be significantly smaller compared to the overall droplet volume. Our estimate suggests that the thickness of the surface region with enhanced ice nucleation frequency is ~2 nm. Consequently, when

estimating ice nucleation frequencies per unit volume, observing the pronounced impact of free-surface-induced ice nucleation becomes challenging. There have been noteworthy experiments investigating ice nucleation rates in nanodroplets[48–50]. However, comparing these rates to those in bulk poses challenges, primarily due to the difficulty of accessing the bulk nucleation rate, coupled with the combined influence of surface effects and the Laplace pressure effect. Nevertheless, it is crucial to emphasise that experimental confirmation regarding the facilitation of ice nucleation by the free surface holds profound significance within the realm of fundamental scientific understanding.

In this context, our study provides compelling evidence for the surface-induced enhancement of ice nucleation in the mW water model, albeit with a relatively modest degree of enhancement. Moreover, we also observe robust support for surface-induced freezing in the realistic water model, specifically the TIP4P/ice model. To establish a more precise connection with experimental data, it is highly desirable to investigate the ice nucleation rate in thin films using a realistic water model, employing a robust method such as brute-force simulations.

Here, we note that the impact of surface-induced negative pressure on liquid structures and the resulting enhancement of order formation might have analogous effects in various materials, especially tetrahedral liquids like silicon, germanium, carbon, and silica. Notably, previous studies have reported similar free-surface-induced crystallisation in silicon[30] and the Stillinger-Weber potential with $\lambda = 21$[51]. This raises an intriguing question regarding whether the surface-induced crystallisation in these systems shares a common mechanism with water or not.

Lastly, this study emphasises the importance of carefully choosing order parameters in biased simulations, commonly employed to investigate rare events such as crystal nucleation. The efficacy of these simulations relies on the judicious selection of order parameters that accurately characterise the kinetic pathways. Our findings underscore the critical nature of appropriate order parameter selection in biased simulations, as also discussed in Refs. 28–30.

## Methods

We perform molecular dynamics simulations using a large-scale atomic/molecular massively parallel simulator (LAMMPS) for both mW and TIP4P/ice systems. Bulk water and films&droplets are conducted in the *NPT* and *NVT* ensembles, respectively, with periodic boundary conditions. The Nose-Hoover thermostat and the Parrimello-Rahman barostat are applied to control the temperature and pressure, respectively. The dimensions of the simulation boxes along *x*, *y*, and *z* directions are 9.44 nm, 9.44 nm, and 21 nm, respectively. The time step is 1 fs. 3D local structural order is analysed using the coarse-grained Steinhardt bond orientational order parameter $Q_{12}$[24] and the CHILL algorithm, whereas 2D structural order on the free surface is analysed using quasi-2D order parameter $\bar{\lambda}_1$.

Ice 0 has two different local environments around the central molecule: (5,5,5,5,6,6) and (5,5,5,7,6,6), as shown in Fig. 2e. A molecule is classified as Ice 0-like if at least 5 of the six rings around it with 16 neighbours within cutoff $r_c = 0.488$ nm have the correct number of the Ice 0 ring members. $f_n$ indicates the fraction of the *n*-membered ring around a molecule.

### $Q_{12}$ and connections

To characterise bond orientational order of ice-like structures, we use the coarse-grained order parameter, $Q_{12}$, defined as follows[24],

$$q_{12,m}(i) = \frac{1}{N_b(i)} \sum_{j=1}^{N_b(i)} Y_{12,m}(\widehat{r_{ij}}), \qquad (1)$$

and then the averaged form is,

$$Q_{12,m}(i) = \frac{1}{\widetilde{N_b(i)}} \sum_{k=0}^{\widetilde{N_b(i)}} q_{12,m}(k). \tag{2}$$

The scalar product between $Q_{12,m}$ of two molecules is defined as $S_{ij} = \sum_{k=-12}^{12} Q_{12,m}(i)Q^*_{12,m}(j)$. A pair of molecules $i$ and $j$ is identified as connected if $S_{ij}$ is larger than 0.75. If the number of connections that the molecules have is above 75% $N_b(i)$, the molecule is identified as crystalline.

### The CHILL algorithm

The local order around each water molecule $i$ is defined by a local orientational bond order parameter vector $q_l$ with 2l+1 complex components,

$$q_{lm}(i) = \frac{1}{4} \sum_{j=1}^{4} Y_{l,m}(\widehat{r_{ij}}), \tag{3}$$

where $q_{l,m}(i)$ projects the bond orientational order of the four closest neighbours of a molecule based on spherical harmonics. The alignment of the orientation of the local structures is measured by the normalised dot product of $q_l$ between each pair of neighbour molecules:

$$c_{ij} = \frac{\sum_{m=-l}^{l} q_{lm}(i)q^*_{lm}(j)}{\left(\sum_{m=-l}^{l} q_{lm}(i)q^*_{lm}(i)\right)^{\frac{1}{2}} \left(\sum_{m=-l}^{l} q_{lm}(j)q^*_{lm}(j)\right)^{\frac{1}{2}}} \tag{4}$$

### The quasi-2D ice nucleation

We chose the order parameter $\bar{\lambda}_1(i)$[52] to identify the crystal particles. However, the 2nd layer is not an exact 2D but a quasi-2D with bilayer substructures.

$$\lambda(i) = \frac{1}{N_b(i)} \sum_{j}^{N_b(i)} \left[ \sum_{m \pm 6} \hat{q}_{6m}(i)\hat{q}_{6m}^*(j) \right] \tag{5}$$

$$\bar{\lambda}_1(i) = \frac{1}{1 + N_b(i)(1 + N_b(j))} \left[ \lambda(i) + \sum_{j \in N_b(i)} \left( \lambda(j) + \sum_{k \in N_b(j)} \lambda(k) \right) \right] \tag{6}$$

According to the distribution of $\bar{\lambda}_1(i)$ of interfacial liquid and ice, we can identify the ice molecules in the contact layer as follows.

| | |
|---|---|
| $\bar{\lambda}_1(i) \leq 0.23$ | liquid |
| $0.23 < \bar{\lambda}_1(i) \leq 0.4$ | ice-preorder |
| $\bar{\lambda}_1(i) > 0.4$ | ice |

Then, the ice molecules in the second layer are identified as

| | |
|---|---|
| $\bar{\lambda}_1(i) \leq 0.2$ | liquid |
| $0.2 < \bar{\lambda}_1(i) \leq 0.26$ | ice-preorder |
| $\bar{\lambda}_1(i) > 0.26$ | ice |

### The estimation of the nucleation rate $R$

Here, we determine the waiting time, denoted as the induction time $t_{ind}$, for each trajectory through the time evolution of potential energy

(refer to supplementary Fig. 2). Utilising the distribution of induction times for nucleation, we estimate the probability for a single simulation trajectory to remain in the liquid state at time $t$ using the expression:

$$P_{liq}(t) = 1 - \frac{1}{N_{sim}} \sum_{i=1}^{N_{sim}} \Theta(t - t_{ind}^i), \tag{7}$$

where $N_{sim}$ represents the total number of simulations conducted for each system (in this case, 50), $t_{ind}^i$ is the induction time determined for the $i$-th simulation, and $\Theta(x)$ is the Heaviside step function. Subsequently, applying Eq. (7), we derive $P_{liq}(t)$. To assess the nucleation rate from the shape of $P_{liq}(t)$, we employ a stretched exponential function:

$$P_{liq}(t) = \exp[-(Rt)^\gamma], \tag{8}$$

where $R$ denotes the nucleation rate. The values of $R$ acquired by fitting this function to the $P_{liq}(t)$ data are depicted in Fig. 1d.

## Data availability

All study data are included within the article and Supplementary Information. Source data are provided with this paper.

## Code availability

The simulation codes used in this study are available from the corresponding authors upon request.

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

## Acknowledgements

This work was supported mainly by the "Frost Protection Science" project at the Institute of Industrial Science, the University of Tokyo, which was established as one of the social cooperation programs under the agreement between the University of Tokyo and Daikin Industries, Ltd. The work was also supported partially by Specially Promoted Research (JP20H05619). from the Japan Society of the Promotion of Science (JSPS).

## Author contributions

H.T. designed and supervised the project. G.S. performed research; G.S. and H.T. analysed data and wrote the paper together.

## Competing interests

The authors declare no competing interests.
