## [Peer Review File · Nature Communications]

Surface-induced water crystallization driven by precursors formed in negative pressure regionsReviewers' Comments:

Reviewer #1 (Remarks to the Author):

In this work, the authors carried out a molecular dynamics study of ice nucleation within a freestanding water film. The study was motivated by the controversy regarding the role of water-air surface on ice nucleation. Despite the proposed surface-induced crystallization, there is a clear lack of experimental evidence, and results from molecular modeling are also inclusive. The authors employed the mW water model, which has been used in prior modeling studies that did not find evidence of surface freezing, to conduct MD simulation and reported the preference of formation of Ice 0 structure within the subsurface of water film. The preference was attempted to be rationalized by the authors through the abundance of topological fragments resembling ice 0 near surface induced by negative pressure due to surface tension.

Despite the significance of the topic itself, there are a few major scientific issues that make the paper not suitable for publishing on Nature Communications. The most critical issue is that there is NO clear evidence of surface freezing in this study. The only "evidence" is the formation of ice 0 structure near surface, which is far from being sufficient. Even in the case where water film yields a rate lower than or close to homogeneous rate, there was still a tendency for ice nucleation to occur near surfaces. In particular, the thickness of water film in the study 8 nm is barely enough to differentiate bulk region from surface region. Finite size effect has been previously demonstrated to play an important role in water films. How would the picture change with an increasing film thickness?

More relevant questions should be addressed to claim surface induced ice nucleation: Did water film yield a higher nucleation rate than bulk water of the same volume, or did water film crystallize at a statistically higher temperature? Only a positive answer to either question can constitute a more meaningful evidence of surface crystallization. The authors have already obtained 50 independent trajectories which should allow them to extract unbiased ice nucleation rate (along with its statistical uncertainty) for water film. How does it compare with homogeneous ice nucleation at the same temperature (206 K)? If the rates are indistinguishable, how would one conclude free surface promotes ice nucleation?

Alternatively, they can also investigate the free energy barrier. The authors made a strong claim that the order parameters used in previous studies did not capture the relevant pathway and argued the most relevant order parameter should be Q12. Although this reviewer does not necessarily agree on this, the authors can at least try to make it self-consistent by showing ice nucleation in subsurface indeed leads to a lower free energy barrier using the order parameter based on Q12. They do have all the data to allow them to do so.

Other than these major issues, the overall writing of the paper can be improved. Certain part of the paper is difficult to understand. For example, page 9-10 where the authors discussed the role of Laplace pressure and commented on previous studies. For those who may not know exactly these references, it's rather difficult to understand the main points through these sentences because they are mixed with authors' opinion. By the way, when claiming others' conclusion "is not correct" multiple times in the paper, in what position do the authors believe they have to make such a strong statement? Lastly, when trying to generalize their conclusion to other materials like tetrahedral liquids, the authors should be aware of early studies that already attempted to tackle this problem, for example, Nat. Mater. 8, 726 (2009); JCP 18, 4102 (2016).

Reviewer #2 (Remarks to the Author):

This study reports molecular simulations showing that ice tends to nucleate with greater probability at a certain distance from the 'free surface' of supercooled water, and attributes this to surface induced negative pressure favouring so-called Ice-0 like symmetry. It's an interesting study however I have a few major concerns and would not suggest publication of the manuscript in its current form. I have detailed these concerns below.

Major concerns:

A central claim of the paper is that ice nucleation is favoured some distance, roughly 2 nm, from the water surface. I have a few issues with this. Firstly, it is not obvious to me that what is shown in Fig. 1 (b) is necessarily statistically significant. I would suggest the authors find some way of comparing their results to the null hypothesis that nucleation takes place uniformly across the box of water. The asymmetry of the peaks at roughly 2 and -2 nm, suggests that the finding may not be robust.

I find the idea of negative pressure in a droplet or film a bit slippery. Do you not need to consider the direction of pressure? This said, wouldn't you expect the negative pressure due to surface tension to be at a maximum closer to the surface than 2 nm? See Sega et al. (J. Chem. Theory Comput. 2016, 12, 9, 4509–4515). Similarly, if the claim is that enhancement in nucleation rate is due to negative pressure wouldn't it make sense to work out a pressure profile for the system and compare it to the cluster distribution profile across the system?

I have a concern about the experimental testability and real-world relevance of the study. The introduction motivates the work on the basis of relevance to various applied problems where ice formation matters yet there is no discussion of how these findings might help us understand real systems. I think the question in the end is 'do we expect a small water droplet, with a water-vapour interface to freeze at a warmer or colder temperature than would be anticipated for an equivalent volume of 'bulk' water?'. The work here would presumably predict that smaller droplets, containing a greater proportion of 'interfacial' water would tend to have a higher nucleation rate per volume than larger droplets? To the best of my knowledge, there is no experimental evidence for surface enhancement of apparently homogeneous ice nucleation (Atkinson et al. J. Phys. Chem. A 2016, 120, 33, 6513–6520 and Duft and Leisner Atmos. Chem. Phys., 2004, 4, 1997–2000). I would note here that some of the work cited (Shaw et al. J. Phys. Chem. B 2005, 109, 9865–9868) where nucleation is enhanced near a surface refers to heterogeneous ice nucleation and would seem to me to be of limited relevance to the present study.

It is worth noting that Hayton et al. (DOI: 10.1039/D3FD00099K (Paper) Faraday Discuss., 2023) have very recently attempted to assess how we might expect the thinness of water films to impact their ice nucleation rate compared to a more naively calculated homogeneous ice nucleation rate and found that the difference is minimal. An approach of this type might help clarify the relevance of the present study.

Finally, I do not think that the importance of identifying ice-0 like precursors is sufficiently well articulated at present. I am all for fundamental science, but I think it important to lay out how knowing the structure of critical cluster in a given situation might have wider scientific impact.

Reviewer #3 (Remarks to the Author):

This manuscript investigates ice nucleation by means of molecular dynamics simulations. From a methodological standpoint, the authors study ice nucleation in water films and droplets, make use of two water models, namely mW and TIP4P/ice, and also put the spotlight on the relevance of choosing suitable order parameters.

The main result of the work consists in revealing a novel nucleation mechanism in which the free surface-induced negative pressure promotes nucleation via the emergence of Ice 0-like precursors with ring topologies containing 5 and 6-membered rings. Since these precursors are expected to be more friendly to liquid water as compared to standard (bulk) ice-like topologies, they are expected to lower the interfacial energy cost for ice nucleation. This finding would rectify previous statements regarding free surface order as "bad" for ice nucleation. Additionally, unlike substrate-induced ice nucleation that occurs directly at the outermost layer, here the process occurs slightly inwards (at the sub-surface layer, but very close to the surface). Hence, the authors not only reproduce the surface-induced crystallisation observed in experiments, but they also unravel its physical molecular underpinnings.

From the above-explained considerations I find this work to tackle a problem of utmost relevance, both from fundamental and practical perspectives, and to present a solid, comprehensive and carefully described approach that yields highly significant results. More importantly, the study provides a new conceptualization that proposes a solution to the core physical and molecular basis of the problem. Thus, I am glad to recommend publication.

There is only one point I would like the authors to comment. The local orientational bond order, $q_{lm}(i)$, is calculated by considering the four closest neighbours of a molecule. For the molecules at the first (outermost) layer, where important 3-fold hydrogen bond coordination is expected (with the presence of dangling bonds), the fourth neighbor would lie at the second coordination shell instead of belonging to the first-shell. Thus, unlike the quasi-2D lambda index or the study of five or six-membered rings, this index might produce artifacts in such case. Please comment on the application of this metric in your study.

Replies to the comments of Reviewer #1:

In this work, the authors carried out a molecular dynamics study of ice nucleation within a freestanding water film. The study was motivated by the controversy regarding the role of water-air surface on ice nucleation. Despite the proposed surface-induced crystallization, there is a clear lack of experimental evidence, and results from molecular modeling are also inclusive. The authors employed the mW water model, which has been used in prior modeling studies that did not find evidence of surface freezing, to conduct MD simulation and reported the preference of formation of Ice 0 structure within the subsurface of water film. The preference was attempted to be rationalized by the authors through the abundance of topological fragments resembling ice 0 near surface induced by negative pressure due to surface tension.

First, we thank the reviewer for carefully reading our manuscript and providing comments. The feedback and guidance provided by the reviewer have been invaluable in enhancing the quality and credibility of our research.

Before going to the specific details, we would like to emphasise that our simulations represent brute-force investigations of direct ice nucleation processes. Our findings consistently demonstrate that ice formations always occur in proximity to the free surface, regardless of film thickness or geometry, encompassing thin films and nanodroplets. It is important to note that we do not employ any enhanced sampling techniques or seeding methods in our simulations. As a result, our conclusion that ice formations tend to occur near the surface rather than in the central region is based on empirical observations and does not depend on any underlying assumptions, including the presence of ice 0. We believe that this strengthens the validity and reliability of our findings.

We also directly measure the nucleation rates of water films and bulk. Nucleation is intrinsically a stochastic process; therefore, even for the same system, we must wait for different time intervals to observe nucleation. Thus, to estimate the nucleation rate R , we performed more than 50 simulations for each system under the same condition. Subsequently, we determined the crystal birth time, i.e., the induction time for crystal nucleation (t_{ind}), for each trajectory by the time evolution of the potential energy (see Fig. R1a). Based on the distribution of the induction times of the nucleation, we could estimate the probability for a single simulation trajectory to stay in the liquid state at time t :

$$P_{liq}(t) = 1 - \frac{1}{N_{sim}} \sum_{i=1}^{N_{sim}} \Theta(t - t_{ind}^i), \quad (1)$$

where N_{sim} is the total number of simulations performed for each system, t_{ind}^i is the induction time determined for the i -th simulation, and $\Theta(x)$ is the Heaviside step function. Then, following Eq. (1), we calculated $P_{\text{liq}}(t)$. Examples of $P_{\text{liq}}(t)$ for various conditions are presented in Fig. R1b.

By examining the shape of $P_{\text{liq}}(t)$, we can quantify the nucleation rate by fitting the stretched exponential function to $P_{\text{liq}}(t)$:

$$P_{\text{liq}}(t) = \exp[-(Rt)^\gamma] \quad (2)$$

where R is the nucleation rate. The fitting results are displayed in Fig. R1b. To compare the nucleation rates of bulk water and water films, we illustrate the nucleation rate per volume, denoted as R/V ($\text{nm}^{-3}\text{ns}^{-1}$), in Fig. R1c. It is evident from the results that the nucleation rate of the water film is higher than that of the bulk at different temperatures, and furthermore, it increases as the water film becomes thinner.

Figure R1. Estimation of the nucleation behaviour. The nucleation rates were directly measured by tracking the time evolution of potential energy for both bulk water and water films with thicknesses of $L=8$ nm, 6 nm and 4 nm at $T=200$ K and 206 K. **a**, The determination of the induction time (t_{ind}) by monitoring the time evolution of the potential energy. **b**, The probability of a system remaining in a liquid state as a function of time, $P_{\text{liq}}(t)$, for different conditions. **c**, The comparison of the nucleation rate per volume R/V between bulk water and water films with different thicknesses.

In the following, we provide individual responses to each of the reviewer's comments.

Despite the significance of the topic itself, there are a few major scientific issues that make the paper not suitable for publishing on Nature Communications. The most critical issue is that there is NO clear evidence of surface freezing in this study. The only “evidence” is the formation of ice 0 structure near surface, which is far from being sufficient. Even in the case where water film yields a rate lower than or close to homogeneous rate, there was still a tendency for ice nucleation to occur near surfaces. In particular, the thickness of water film in the study 8 nm is barely enough to differentiate bulk region from surface region. Finite size

effect has been previously demonstrated to play an important role in water films. How would the picture change with an increasing film thickness?

We respectfully disagree with the reviewer's assessment of the points raised. We are afraid that there might be some misunderstanding. The reviewer specifically noted our observation of “ice 0”. However, it is crucial to understand that metastable “ice 0”-like structures only exist transiently in the early stage of crystal nucleation. Ice 0-like structures formed near the free surface quickly transform into standard ice crystals, such as cubic ice. Thus, our observation primarily pertains to ice nucleation near the surface, a fact substantiated by the clear peak evident in the probability distribution function (see new Fig. 1b), which indicates a heightened likelihood of ice nucleation occurring near the surface.

It is also important to highlight that the identification of ice nuclei is directly carried out using the bond orientational order parameter Q_{12} , rather than “Ice 0”-like structures. Additionally, the direct measurement of the nucleation rate for both bulk and water films, as illustrated in Fig. R1 above, serves to confirm our observation regarding nucleation induced by the free surface.

We want to emphasise that our conclusion is drawn from direct observations in brute force simulations, eliminating reliance on assumptions. Our findings are rooted in thorough and extensive computational investigations, ensuring the reliability and robustness of our results.

While it is true that our system size may be comparatively small, it is important to acknowledge that this size is widely accepted as standard practice within the field of computer simulations focused on ice nucleation (see, e.g., references [2,3] below). Notably, even renowned research groups led by world experts such as Professor Pablo Debenedetti and Professors Angelos Michaelides and Stephen J. Cox employed similar simulation setups in their investigations, with film thicknesses of 4 nm in the former [3] and ranging from 2.5 to 6 nm in the latter [2]. Moreover, we confirmed our conclusion by using a droplet with a diameter of 12 nm. It is worth highlighting that our film thickness, at 8 nm, exceeds those utilised in these seminal works.

The findings of both groups supported the notion that ice nucleation is more likely to occur in the central region of a thin water film. We attribute the discrepancy between the conclusion of Professor Debenedetti's group and ours to a critical factor - the selection of bond-orientational order parameters. We have elaborated on this aspect in detail in our manuscript, explaining how the choice of these parameters may have led to differing conclusions between our study and Professor Debenedetti's work.

On the other hand, the differences in the conclusions between our study and the research conducted by Professors Michaelides and Cox can be attributed to the utilisation of a seeding technique. Specifically, they employed a method that involved seeding a spherical nucleus in the middle of the film [3], a choice that may not be appropriate.

Here, we briefly discuss a potential problem with the seeding method. First, the interfacial energy cost to form the initial ice nucleus is not associated with standard ice structures, such as cubic or hexagonal ices, but rather with ice 0-like structures at the interface, as discussed in our previous paper [4] (we note that the existence and metastability of the ice 0 crystal was confirmed through density functional theory and quantum Monte Carlo simulations in Ref. [5]). When introducing a seed into a liquid, the surface structures of the seed should rapidly equilibrate, spontaneously forming ice 0-like structures on its surface if these structures are relevant. Consequently, the nucleation frequency in bulk should be appropriately estimated using the seeding method for bulk in this context.

However, for thin films, it is crucial to recognise that identifying the likely locations of the formation of ice nuclei and understanding their shapes represents a distinct and more complex aspect of the investigation. For example, our study indicates that ice nuclei tend to form in the subsurface regions and not in the middle region. These considerations indicate that the selection of the seed's position and shape may play a crucial role in estimating the nucleation rate in the presence of symmetry-breaking surface fields, unlike in bulk.

[1] Emily B. Moore and Valeria Molinero, Structural transformation in supercooled water controls the crystallization rate of ice. *Nature* **479**, 506 (2011).

[2] John A. Hayton, Michael B. Davies, Thomas F. Whale, Angelos Michaelides, and Stephen J. Cox, The limit of macroscopic homogeneous ice nucleation at the nanoscale, *Faraday Discuss.* (in press).

[3] Amir Haji-Akbari and Pablo G. Debenedetti, Perspective: Surface freezing in water: A nexus of experiments and simulations, *J. Chem. Phys.* **147**, 060901 (2017).

[4] John Russo, Flavio Romano & Hajime Tanaka, New metastable form of ice and its role in the homogeneous crystallization of water, *Nat. Mater.* **13**, 733–739 (2014).

[5] D. Quigley, D. Alfè, and B. Slater, On the stability of ice 0, ice i, and Ih, *J. Chem. Phys. Rapid Communications* **141**, 161102 (2014).

Concerning the finite size effect, we do agree with the reviewer's suggestion that we should examine the finite size effect. We have indeed investigated this aspect by changing the film thickness from 4 nm to 8 nm and can confidently affirm that our conclusions remain unaffected by the finite-size effect. We would like to note using a film thicker than 8 nm becomes computationally impractical due to the substantial computational demands associated with the need for a large number of trajectories. Regardless of the film thickness or the system's geometry (be it flat films or nanodroplets), the consistent outcome is the nucleation of ice crystals at a distance of approximately 2 nanometers from the free surface. It is worth noting that our results, obtained using the mW water model that employs a short-range potential, are not so strongly influenced by finite-size effects. This is in contrast to systems with long-range electrostatic interactions, where such effects can be significant.

We believe that the provided explanations effectively address any concerns the reviewer may have had regarding the reliability of our conclusion.

More relevant questions should be addressed to claim surface induced ice nucleation: Did water film yield a higher nucleation rate than bulk water of the same volume, or did water film crystallize at a statistically higher temperature? Only a positive answer to either question can constitute a more meaningful evidence of surface crystallization. The authors have already obtained 50 independent trajectories which should allow them to extract unbiased ice nucleation rate (along with its statistical uncertainty) for water film. How does it compare with homogeneous ice nucleation at the same temperature (206 K)? If the rates are indistinguishable, how would one conclude free surface promotes ice nucleation?

We would like to express our appreciation to the reviewer for providing valuable comments. Previous computational investigations in this field have largely approached the problem by examining the locations where ice crystals tend to form. However, we acknowledge the merit in the reviewer's suggestion to incorporate a direct comparison of nucleation frequencies between the thin water film and the bulk phase, as it can offer valuable insights into the phenomenon.

We conducted direct measurements of the nucleation rate per unit volume for both water films and bulk (refer to Fig. R1 above). The results indicate that at the same temperature, specifically for $T=200$ K and 206 K, the nucleation rate of the water film surpasses that of the bulk. This strong support for the promotion of ice nucleation by the free surface is evident.

Alternatively, they can also investigate the free energy barrier. The authors made a strong claim that the order parameters used in previous studies did not capture the relevant pathway and argued the most relevant order parameter should be Q12. Although this reviewer does not necessarily agree on this, the authors can at least try to make it self-consistent by showing ice nucleation in subsurface indeed leads to a lower free energy barrier using the order parameter based on Q12. They do have all the data to allow them to do so.

We thank the reviewer for the valuable comment. Indeed, estimating the free energy barrier for crystal nucleation is an excellent approach to further validate our argument. However, as previously mentioned, we have already conducted the method suggested by the reviewer in the earlier comment and found that it supports the validity of our argument. Thus, we would like to leave this for future investigation.

Regarding the use of Q12 order parameters, we have conducted extensive research that demonstrates their effectiveness in detecting various crystal structures, including ice 0. We previously made the discovery of a new metastable ice, named “ice 0”, and documented it in reference [4]. Subsequently, its existence and stability were confirmed through density functional theory and quantum Monte Carlo simulations in reference [5]. In our paper [4], we unveiled that ice-0-like structures serve as precursors or catalysts for the nucleation of cubic ice in bulk. In our current study, our findings demonstrate that these ice-0-like structures are preferentially generated near the free surface and play a pivotal role in promoting ice nucleation in proximity to that surface.

In reference [4], we also emphasised the critical importance of utilising the Q12 bond-orientational order (BOO) parameter when studying tetrahedral liquids. It is worth noting that the conventional Q6 parameter is unable to detect certain significant crystal structures, including metastable ice 0. For example, by employing the Q12 parameter, we not only uncovered the existence of the ice 0 crystal in water but also made a discovery of a new crystalline form in silicon. This form was later identified as the β -Sn crystal structure through a straightforward atom displacement and a reduction in cell symmetry, as described in reference [7]. These findings underscore the paramount importance of incorporating the Q12 order parameter into investigations of crystallisation in tetrahedral liquids.

Furthermore, it is essential to highlight that using the Q6 parameter in biased Monte Carlo or forward-flux sampling simulations can potentially lead to the oversight of ice nucleation processes involving ice 0-like structures. We believe that this oversight may have contributed to the contrasting conclusions reached in previous computer simulations compared to our own research.

[6] Flavio Romano, John Russo, and Hajime Tanaka, Novel stable crystalline phase for the Stillinger-Weber potential, *Phys. Rev. B* 90, 014204 (2014).

[7] Domagoj Fijan, Mark Wilson, The characterisation of the “X” crystal structure in the Stillinger-Weber potential, *Chem. Phys. Lett.* 685, 316–321 (2017).

Other than these major issues, the overall writing of the paper can be improved. Certain part of the paper is difficult to understand. For example, page 9-10 where the authors discussed the role of Laplace pressure and commented on previous studies. For those who may not know exactly these references, it’s rather difficult to understand the main points through these sentences because they are mixed with authors’ opinion. By the way, when claiming others’ conclusion “is not correct” multiple times in the paper, in what position do the authors believe they have to make such a strong statement? Lastly, when trying to generalize their conclusion to other materials like tetrahedral liquids, the authors should be aware of early studies that already attempted to tackle this problem, for example, *Nat. Mater.* 8, 726 (2009); *JCP* 18, 4102 (2016).

We thank the reviewer for his/her kind and valuable comments. Following suggestions, we have reorganised and improved the sections. Furthermore, we have incorporated the references mentioned by the reviewer into our revised manuscript to enhance the credibility and completeness of our work.

We trust that our revised manuscript represents a substantial improvement, with our conclusions now being more robustly supported. We sincerely hope that the reviewer finds our revised work well-suited for publication in *Nature Communications*.

Replies to the comments of Reviewer #2:

This study reports molecular simulations showing that ice tends to nucleate with greater probability at a certain distance from the ‘free surface’ of supercooled water, and attributes this to surface induced negative pressure favouring so-called Ice-0 like symmetry. It’s an interesting study however I have a few major concerns and would not suggest publication of the manuscript in its current form. I have detailed these concerns below.

First, we thank the reviewer for carefully reading our manuscript and providing comments. We are pleased to learn that the reviewer found our study interesting. The feedback and guidance offered by the reviewer have been instrumental in elevating the quality and credibility of our research.

Before going to the specific details, we would like to emphasise that our simulations represent brute-force investigations of direct ice nucleation processes. Our findings consistently demonstrate that ice formations always occur in proximity to the free surface, regardless of film thickness or geometry, encompassing thin films and nanodroplets. It is important to note that we do not employ any enhanced sampling techniques or seeding methods in our simulations. As a result, our conclusion that ice formations tend to occur near the surface rather than in the central region is based on empirical observations and does not depend on any underlying assumptions, including the presence of ice 0-like structures. We believe that this strengthens the validity and reliability of our findings.

In the following, we provide individual responses to each of the reviewer's comments.

Major concerns:

A central claim of the paper is that ice nucleation is favoured some distance, roughly 2 nm, from the water surface. I have a few issues with this. Firstly, it is not obvious to me that what is shown in Fig. 1 (b) is necessarily statistically significant. I would suggest the authors find some way of comparing their results to the null hypothesis that nucleation takes place uniformly across the box of water. The asymmetry of the peaks at roughly 2 and -2 nm, suggests that the finding may not be robust.

We appreciate the reviewer's valuable, constructive comments. We fully agree with the reviewer's observation that the double peak shown in Fig. 1(b) should be symmetric if our proposed mechanism is correct. However, we would like to clarify that the asymmetry in the distribution is primarily attributed to limited sampling. To address this issue, we have significantly increased the number of

samples, resulting in a more symmetric distribution. We believe this improvement will remove the reviewer's concerns regarding the validity of our conclusion.

Furthermore, we have undertaken a comparison of ice nucleation frequencies between thin films and the bulk, as suggested by Reviewer #1. The results are presented in Fig. R1 above in this response letter (new Fig. 1c in the manuscript). Our findings unequivocally confirm that the ice nucleation frequency is indeed higher in thin films than in the bulk.

We believe that these results will effectively address any concerns raised by the reviewer and substantiate the validity of our conclusion.

I find the idea of negative pressure in a droplet or film a bit slippery. Do you not need to consider the direction of pressure? This said, wouldn't you expect the negative pressure due to surface tension to be at a maximum closer to the surface than 2 nm? See Sega et al. (J. Chem. Theory Comput. 2016, 12, 9, 4509–4515). Similarly, if the claim is that enhancement in nucleation rate is due to negative pressure wouldn't it make sense to work out a pressure profile for the system and compare it to the cluster distribution profile across the system?

We appreciate the thoughtful comment from the reviewer. Firstly, we need to consider the direction of pressure. In Fig. 2 in the manuscript, it is evident that the lateral direction displays significant negative pressure instead of the normal direction. This observation aligns with the findings in the work of Amir and Pablo [1]. Additionally, we consistently observe different profiles along the radial direction for the normal pressure and tangential pressure of a water droplet in Fig. 3e and Fig. 5 in Shahraza et al.'s study [2]. Specifically, it is the tangential direction that manifests negative pressure in a water droplet, with the minimum of the negative pressure occurring approximately 0.4–0.5nm from the surface.

According to our results and Sega's findings [3], the normal pressure P_N remains nearly constant throughout the system for both flat and spherical surfaces, ensuring mechanical stability. The difference between the normal (P_N) and lateral (P_T) components of the pressure tensor is crucial for interfaces, as it characterises the surface tension: $\gamma = P_N - P_T$. It is the negative lateral and tangential pressure P_T that contributes to a high surface tension in proximity to the surface. our results are in line with Sega's observations. Moreover, it is the negative pressure in lateral (for flat surface) and tangential (for droplet) directions that plays an important role in lowering the free energy barrier for nucleation. In other words, nucleation is facilitated in regions with negative pressure, and the minimum of the negative pressure aligns well with the peak position of the Ice 0 density profile.

However, pinpointing small nuclei precisely at the position corresponding to the minimum of the negative pressure is not necessary due to the substantial dynamic fluctuations of interfaces (see Fig.1a in the manuscript), i.e., capillary waves. As the ice nuclei grow to a sufficiently large size, we can observe a pronounced preferential position for stable ice nuclei in thermodynamics (see new Fig. 1b). However, since crystal nuclei formed near the surfaces can only grow inwards, the peak positions of stable ice nuclei ($N_c > 60$ molecules) are inevitably observed to be shifted inwards compared to the minimum position of the negative pressure.

Finally, in response to the reviewer's suggestion, we have conducted an in-depth examination of the correlation between the pressure profile and the profiles of pre-ordered structures, i.e., the Ice 0-like clusters. Our findings reveal a close correlation between the two profiles, as illustrated in the newly added Figure 3e in the revised manuscript. We believe that this additional analysis will help to convince the reviewer of the validity of our argument. In fact, Shahraza et al. have assessed the pressure profile of water nanodroplets and clearly show a negative tangential pressure region close to the free surface [3], which strongly supports our findings.

[1] Amir Haji-Akbari and Pablo G. Debenedetti, *Proc. Natl. Acad. Sci.* 114, 3316–3321 (2017).

[2] Shahrazad M A Malek, Francesco Sciortino, Peter H Poole and Ivan Saika-Voivod, *J. Phys.: Condens. Matter* 30, 144005 (2018).

[3] Marcello Sega, Balazs Fabian and Pal Jedlovsky, *J. Chem. Theory Comput.* 12, 4509–4515 (2016).

I have a concern about the experimental testability and real-world relevance of the study. The introduction motivates the work of the basis of relevance to various applied problems where ice formation matters yet there is no discussion of how these findings might help us understand real systems. I think the question in the end is ‘do we expect a small water droplet, with a water-vapour interface to freeze at a warmer or colder temperature than would be anticipated for an equivalent volume of ‘bulk’ water?’. The work here would presumably predict that smaller droplets, containing a greater proportion of ‘interfacial’ water would tend to have a higher nucleation rate per volume than larger droplets? To the best of my knowledge, there is no experimental evidence for surface enhancement of apparently homogeneous ice nucleation (Atkinson et al. *J. Phys. Chem. A* 2016, 120, 33, 6513–6520 and Duft and Leisner *Atmos. Chem. Phys.*, 2004, 4, 1997–2000). I would note here that some of the work cited (Shaw et al. *J. Phys. Chem. B* 2005109, 9865–9868) where nucleation is enhanced near a surface refers to heterogeneous ice nucleation and would seem to me to be of limited relevance to the present

study.

We thank the reviewer for providing valuable comments and for sharing information regarding related papers. As discussed earlier, we conducted a comparison of ice nucleation frequencies between thin water films and bulk water in Fig. R1 above, and our findings indicate that the former exhibits a higher nucleation rate than the latter.

Regarding the experimental papers mentioned by the reviewer, the paper by Atkinson et al. studied water droplets immersed in an oil. Thus, it is not relevant to our research.

On the other hand, the paper by Duft and Leisner compared the freezing probability for droplets of radius 49 μm and 19 μm . They concluded that homogeneous freezing is a volume-proportional process and that surface nucleation might only be important, if at all, for much smaller droplets. As recognized by these authors, the droplet size in this study exceeds ten microns. In such relatively large droplets, the volume of the thin surface region is significantly smaller compared to the overall droplet volume. Consequently, when estimating ice nucleation frequencies per unit volume, it becomes challenging to observe the pronounced impact of free-surface-induced ice nucleation.

Nonetheless, it is of paramount importance to underscore that the discovery regarding the facilitation of ice nucleation by the free surface holds profound significance within the realm of fundamental scientific understanding. In a previous study by Duft and Leisner, it was mentioned in the discussion section that, "It cannot be ruled out, however, that surface nucleation is important for much smaller droplets below 1 μm ."

We hope that our findings will stimulate and encourage further experimental investigations in this direction. It is worth noting, however, that for very small droplets, we must consider the influence of Laplace pressure. As is known, the surface curvature of a water droplet creates an additional pressure called the Laplace pressure, $P_L = P_N$ (normal pressure) - P_T (tangential pressure), which contributes to an increase in internal pressure. As the droplet size decreases, the internal pressure increases (P_L), increases, potentially suppressing the probability of nucleation events in the droplet, as shown in Li et al.'s work [4]. We have included this discussion into the revised manuscript.

In fact, by using the surface tension measurements in experiments, X.Z. Wu et al. [5] discovered surface freezing in liquid normal alkanes 30 years ago. They reported that the freezing temperature of liquid alkanes' surface is up to 3°C above the bulk freezing point. Although water's surface is different from that of alkanes, this experimental work on the surface freezing of liquid alkanes

strongly suggests that conclusions regarding the promotion or suppression of surface freezing in liquids should be approached with careful consideration, rather than relying solely on a few predictions from enhanced sampling methods. Through thorough and direct brute-force simulations of the ice nucleation processes, we robustly provide computational evidence of surface freezing in water. This discovery not only resolves the long-standing controversy in computational studies on the effects of free surfaces on ice nucleation but also holds significant value for further exploration in experiments, emphasising the importance of our work.

[4] Tianshu Li, Davide Donadio and Giulia Galli, Ice nucleation at the nanoscale probes no man's land of water. *Nat. Commun.* **4**, 1887 (2013).

[5] X.Z. Wu, B.M. Ocko, E.B. Sirota, S.K. Sinha, M. Deutsch, B.H.Cao and M.W. Kim, Surface Tension Measurements of Surface Freezing in Liquid Normal Alkanes. *Science* **261**, 1018 (1993).

It is worth noting that Hayton et al. (DOI: 10.1039/D3FD00099K (Paper) Faraday Discuss., 2023) have very recently attempted to assess how we might expect the thinness of water films to impact their ice nucleation rate compared to a more naively calculated homogeneous ice nucleation rate and found that the difference is minimal. An approach of this type might help clarify the relevance of the present study.

We are grateful to the reviewer for bringing to our attention the noteworthy paper by Hayton et al., which we were not aware of, as it has been accepted in Faraday Discussions but has not yet been published. In this paper, the authors stated in the Introduction, “The broad consensus from multiple simulation studies is that ice formation occurs away from the interface, in regions of the fluid that are bulk-like.” This consensus that has emerged within the scientific community stands in stark contrast to the findings we have uncovered in our study.

As emphasised at the beginning of our response, our simulations represent brute-force investigations of direct ice nucleation processes. Our findings consistently demonstrate that ice formations consistently occur in proximity to the free surface, regardless of film thickness or geometry, encompassing thin films and nanodroplets. It is important to note that we do not employ any enhanced sampling techniques or seeding methods in our simulations. As a result, our conclusion that ice formations tend to occur near the surface rather than in the central region is based on empirical observations and does not depend on any underlying assumptions, including the presence of ice 0-like structures. This strengthens the validity and reliability of our findings.

The use of enhanced sampling techniques is warranted only when employing appropriate order parameters for biasing, as we have emphasised in our manuscript. On the other hand, seeding techniques may prove valuable for examining crystal nucleation behaviours. However, it is essential to recognize that ice nucleation near a free surface follows a distinct kinetic pathway involving ice 0-like precursors. Moreover, the selection of the seed's position and shape may also play a crucial role in estimating the nucleation rate in the presence of symmetry-breaking surface fields, unlike in bulk. Consequently, the utility of seeding techniques to investigate the influence of the free surface on ice nucleation is questionable.

To illustrate this point, let's consider the interfacial energy cost in the initial ice nuclei. This cost is not associated with standard ice structures, such as cubic or hexagonal ices, but rather with ice 0-like structures at the interface, as extensively discussed in our previous paper [6]. Additionally, we note that the existence and stability of the ice 0 crystal were further validated through density functional theory and quantum Monte Carlo simulations in reference [7].

Based on these findings, it is important to acknowledge that when introducing a seed into a liquid, the surface structures of the seed should rapidly equilibrate, leading to the spontaneous formation of an ice-0-like interface if it is relevant. Consequently, the nucleation frequency in bulk can be appropriately estimated using the seeding method for bulk in this context. However, it is crucial to recognise that identifying the likely locations of ice nuclei and understanding their shapes represents a distinct and more complex aspect of the investigation. This helps to elucidate why the seeding technique yields a conclusion that is contrary to our own findings.

[6] John Russo, Flavio Romano & Hajime Tanaka, New metastable form of ice and its role in the homogeneous crystallization of water, *Nat. Mater.* 13, 733–739 (2014).

[7] D. Quigley, D. Alfè, and B. Slater, On the stability of ice 0, ice i, and Ih, *J. Chem. Phys. Rapid Communications* 141, 161102 (2014).

Furthermore, it is worth emphasising that our observation of ice nucleation in regions enriched with five-membered rings remains unexplainable if we were to assume that the initial ice nuclei are standard ice structures. This is primarily due to the intrinsic incompatibility of standard ice structures with the presence of five-membered rings.

We hope that the above explanations remove the reviewer's concerns.

Finally, I do not think that the importance of identifying ice-0 like precursors is sufficiently well articulated at present. I am all for fundamental science, but I think it important to lay out how knowing the structure of critical cluster in a given situation might have wider scientific impact.

The reviewer's first comment seems to focus on our observation of "ice 0". However, it is crucial to understand that metastable "ice 0" only exists transiently in the early stage of crystal nucleation. Ice 0-like structures formed near the free surface quickly transform into standard ice crystals, such as cubic ices. More importantly, as emphasized earlier, our conclusion, derived from brute-force simulations, that the free surface assists ice nucleation does not rely on the presence of ice 0. Thus, our observation primarily pertains to ice nucleation near the surface.

We fully agree with the reviewer's opinion that it is critical how our finding has a broad scientific impact. In the paper by Duft and Leisner, it was mentioned in the discussion section that, "It cannot be ruled out, however, that surface nucleation is important for much smaller droplets below 1 μm ." We hope that our findings will stimulate and encourage further experimental investigations in this direction. It is worth noting, however, that for very small droplets, we must consider the influence of Laplace pressure.

We believe that our revised manuscript represents a substantial improvement, with our conclusions now being more robustly supported. This discovery also resolves the long-standing controversy in computational studies on the effects of free surfaces on ice nucleation. We also trust that our findings will have a broad impact on various fields related to ice nucleation. We sincerely hope that the reviewer finds our revised work well-suited for publication in Nature Communications.

Replies to the comments of Reviewer #3:

This manuscript investigates ice nucleation by means of molecular dynamics simulations. From a methodological standpoint, the authors study ice nucleation in water films and droplets, make use of two water models, namely mW and TIP4P/ice, and also put the spotlight on the relevance of choosing suitable order parameters.

The main result of the work consists in revealing a novel nucleation mechanism in which the free surface-induced negative pressure promotes nucleation via the emergence of Ice 0-like precursors with ring topologies containing 5 and 6-membered rings. Since these precursors are expected to be more friendly to liquid water as compared to standard (bulk) ice-like topologies, they are expected to lower the interfacial energy cost for ice nucleation. This finding would rectify previous statements regarding free surface order as “bad” for ice nucleation. Additionally, unlike substrate-induced ice nucleation that occurs directly at the outermost layer, here the process occurs slightly inwards (at the sub-surface layer, but very close to the surface). Hence, the authors not only reproduce the surface-induced crystallisation observed in experiments, but they also unravel its physical molecular underpinnings.

From the above-expounded considerations I find this work to tackle a problem of utmost relevance, both from fundamental and practical perspectives, and to present a solid, comprehensive and carefully described approach that yields highly significant results. More importantly, the study provides a new conceptualization that proposes a solution to the core physical and molecular basis of the problem. Thus, I am glad to recommend publication.

We thank the reviewer for carefully reading our manuscript and valuable comments. We are delighted to receive such a positive evaluation of our manuscript and are pleased to learn that the reviewer has recommended publication with some revisions.

There is only one point I would like the authors to comment. The local orientational bond order, $q_{lm}(i)$, is calculated by considering the four closest neighbours of a molecule. For the molecules at the first (outermost) layer, where important 3-fold hydrogen bond coordination is expected (with the presence of dangling bonds), the fourth neighbor would lie at the second coordination shell instead of belonging to the first-shell. Thus, unlike the quasi-2D lambda index or the study of five or six-membered rings, this index might produce artifacts in such case. Please comment on the application of this metric in your study.

We appreciate the valuable comment from the reviewer. Due to the layering phenomenon in the first two layers, the consideration of neighbours or bond coordination becomes crucial. The values of $q_{lm}(i)$ with the four closest neighbours for the first layer may deviate from the standard. Accordingly, we have introduced new criteria for $q_{lm}(i)$ for ice (Ic/Ih) to identify ice nuclei. Additionally, to avoid the influence of the number of neighbours, we applied several different order parameters, including the CHILL algorithm, quasi-2D lambda, and ring analysis, to fortify the robustness of our conclusion.

To address the reviewer's concern, we have incorporated considerations for the specificity of the free surface in our calculations of both the local bond-order parameters and their coarse-grained counterparts. Comprehensive details about these modifications have been included in the Methods section. We believe that this addition will offer compelling evidence of our meticulous analysis of liquid structures near the surface.

We sincerely hope that the reviewer finds our revised work well-suited for publication in Nature Communications.

REVIEWER COMMENTS

Reviewer #1 (Remarks to the Author):

In this revision, the authors attempted to address all the raised questions. The most significant addition to the original manuscript is the calculated nucleation rates for water films based on the brute-force MD trajectories of spontaneous crystallization. The obtained ice nucleation rates for the films were found to increase as film gets thinner and they were shown to be greater than homogeneous ice nucleation rate (but only by a margin). This new analysis certainly supports the conclusion. However there are two major issues that need to be addressed.

The first issue is an easy one: what is the statistical uncertainty for the estimated rate constant. Is the increase in the nucleation rate statistically meaningful?

The second issue is more serious: I find it rather difficult to reconcile the discrepancy between the current work and prior studies using the same force field. All the previous studies, e.g., Ref. 35, Ref. 14, and Ref. 26, showed that water films led to a decrease of nucleation rate, which is the opposite to what's found in this work. The authors attributed the difference to the application of advanced sampling methods and the choice of order parameter in the prior studies. Although this may be a possibility, the current study along with its analysis isn't convincing enough to prove the point. First, the applicability of advanced sampling approaches in homogeneous ice nucleation based on mW has been cross validated against each other and also against brute-force simulation at low temperature. Second, the current work employs brute-force simulation, which is supposed to be immune to any potential bias in advanced sampling method, but the enhancement due to free surface appears to be very subtle at low temperature, less than a factor of two between bulk and a 4 nm film. How relevant is this change in a more meaningful (much higher) temperature? It is likely this difference will become more significant as temperature increases, but this only stays as a speculation without an explicit demonstration that water film yields a sizable increase (orders of magnitude) in nucleation rate. Third, the authors also blamed on the q_6 -based order parameter for not capturing the most relevant nucleation event. Again, if a Q_{12} order parameter is used in any of these advanced sampling methods, will it yield a higher nucleation rate in film geometry? I think the explicit answer to this question is required to justify a strong claim that overrules the prior multiple studies from multiple groups. Lastly, the enhancement of ice nucleation was attributed to the fraction of ice-0 increasing with negative pressure. I find this argument to be redundant. Negative pressure has already been shown to enhance homogeneous ice nucleation by prior studies, without the necessity to incur ice-0, see for example, Rosky et. al. Chem Phys Lett 789, 139289 (2022). In that study it was also shown that forward flux sampling correctly captures homogeneous ice nucleation rate and the rate enhancement with the negative pressure. Therefore it is rather counterintuitive that the same method would fail to describe rate enhancement in water thin film if there is a such thing and the enhancement is indeed due to negative pressure. I think if the authors make a strong claim against prior findings, all these questions ought to be addressed explicitly.

Reviewer #3 (Remarks to the Author):

I am satisfied with the author's response and, thus, recommend publication.

Responses to the comments of Reviewer #1

In this revision, the authors attempted to address all the raised questions. The most significant addition to the original manuscript is the calculated nucleation rates for water films based on the brute-force MD trajectories of spontaneous crystallization. The obtained ice nucleation rates for the films were found to increase as film gets thinner and they were shown to be greater than homogeneous ice nucleation rate (but only by a margin). This new analysis certainly supports the conclusion. However there are two major issues that need to be addressed.

We would like to express our sincere appreciation to the reviewer for their invaluable feedback on our revised manuscript. Their constructive comments and suggestions from the previous review round have greatly improved the quality of our work. In response to the two issues raised, we will address them individually as follows.

The first issue is an easy one: what is the statistical uncertainty for the estimated rate constant. Is the increase in the nucleation rate statistically meaningful?

In this work, we directly measure the nucleation rate by using the distribution of the induction times of the nucleation,

$$P_{liq}(t) = 1 - \frac{1}{N_{sim}} \sum_{i=1}^{N_{sim}} \Theta(t - t_{ind}^i)$$

where N_{sim} is the total number of simulations performed for each system, t_{ind}^i is the induction time determined for the i -th simulation, and $\Theta(x)$ is the Heaviside step function.

The number of nucleation trajectories does indeed have a slight influence on the shape of $P_{liq}(t)$. However, the distribution of induction times converges when the number of independent statistical simulation trajectories exceeds 50. In our revised manuscript, we conducted 100 independent simulations for each system under identical conditions. Therefore, the statistical uncertainty regarding the estimated rate constant does not impact the conclusions drawn from our investigation into the nucleation behavior of water films.

To confirm the statistical uncertainty, we present the nucleation rates of water films with $L=6$ nm and 4 nm at $T=200$ K and 206 K, respectively, as a function of the number of the statistical simulation trajectories (50~200 trajectory samplings).

Figure R1. The nucleation rate per volume, R/V , as a function of the number of statistical simulations. This figure includes two water films with different thicknesses, $L=6$ nm and 4 nm, at $T=206$ K and 200 K, respectively.

We observe that the statistical uncertainty for the estimated nucleation rate, determined through the distribution of induction times of nucleation, is quite small and can be considered negligible. Hence, the nucleation rate of water films obtained through direct measurement holds statistical significance.

The second issue is more serious: I find it rather difficult to reconcile the discrepancy between the current work and prior studies using the same force field. All the previous studies, e.g., Ref. 35, Ref. 14, and Ref. 26, showed that water films led to a decrease of nucleation rate, which is the opposite to what's found in this work. The authors attributed the difference to the application of advanced sampling methods and the choice of order parameter in the prior studies. Although this may be a possibility, the current study along with its analysis isn't convincing enough to prove the point.

First, the applicability of advanced sampling approaches in homogeneous ice nucleation based on mW has been cross validated against each other and also against brute-force simulation at low temperature. Second, the current work employs brute-force simulation, which is supposed to be immune to any potential bias in advanced sampling method, but the enhancement due to free surface appears to be very subtle at low temperature, less than a factor of two between bulk and a 4 nm film. How relevant is this change in a more meaningful (much higher) temperature? It is likely this difference will become more significant as temperature increases, but this only stays as a speculation without an explicit demonstration that water film yields a sizable increase (orders of magnitude) in nucleation rate. Third, the authors also blamed on the q_6 -based order parameter for not capturing the most relevant nucleation event. Again, if a Q_{12} order parameter is used in any of these advanced sampling methods, will it yield a higher nucleation rate in film geometry? I think the explicit answer to this question is required to justify a strong claim that overrules the prior multiple studies from multiple groups. Lastly, the enhancement of ice nucleation was attributed to the fraction of ice-0 increasing with negative pressure. I find this argument to be redundant. Negative pressure has already been shown to enhance homogeneous ice nucleation by prior studies, without the necessity to incur ice-0, see for example, Rosky et. al. Chem Phys Lett 789, 139289 (2022). In that study it was also shown that forward flux sampling correctly captures homogeneous ice nucleation rate and the rate enhancement with the negative pressure. Therefore it is rather counterintuitive that the same method would fail to describe rate

enhancement in water thin film if there is a such thing and the enhancement is indeed due to negative pressure. I think if the authors make a strong claim against prior findings, all these questions ought to be addressed explicitly.

We appreciate the reviewer's raising this issue. First, we want to clarify that our work does not challenge the validity of advanced sampling approaches in ice nucleation studies. Rather, our point in the manuscript is that these advanced sampling methods, which place intermediate states along the reaction coordinate or an order parameter, may be sensitive to the choice of order parameters.

The local bond orientational order parameter q_6 , commonly used in previous studies (e.g., Ref. 14), fails to adequately describe water nucleation in heterogeneous systems. This could be due to the following reasons:

(1) The local 6th-order harmonic q_6 is not ideal for capturing the correct symmetry of crystals of tetrahedral liquids, including the ice lattice [1,2]. Instead, the 3rd-order (q_3) [1,2] and the 12th-order harmonic (Q_{12}) [3], coarse-grained up to neighboring molecules, are more suitable choices for identifying crystal symmetry for tetrahedral liquids. This is particularly evident in the case of the coarse-grained 12th-order harmonic, Q_{12} (refer to Fig. R3c).

(2) The local order parameter q_6 is measured using the nearest neighbors, which can pick up the ordinary ice structures (hexagonal and cubic ices). In a low-temperature liquid, water molecules and their nearest neighbors tend to form tetrahedral structures to minimize energy. However, to form ice crystalline structures, these tetrahedral geometries need further adjustments to establish specific connections, which involve more neighbors, including at least the second nearest neighbors. Therefore, the local order parameter q_6 generally tends to overidentify the ice molecules in the liquid phase (refer to Fig. R3a).

Figure R2. Illustration of local water geometry and ice structure. (a) Tetrahedral structure in water formed with its nearest neighbors. (b) The ice structure formed by a water molecule and its 16 neighbors through a special tetrahedron connection.

Figure R3. The ice molecules in a water film with $L=8$ nm at $T=206$ K and $t=2.63$ ns. (a) Ice molecules identified using local q_6 as in Refs. 14, 26, and 35. (b) Ice molecules identified using the local q_3 [2]. (c) Ice molecules identified using the coarse-grained Q_{12} [3].

In conclusion, we do not question the validity of advanced sampling methods in liquid films or the various potential models. However, it is important to note that these advanced sampling methods are sensitive to the choice of order parameter. Previous studies have demonstrated that the *local* 6th-order harmonic, q_6 , is not the suitable choice for carbon [1] and silicon [2], as it may overestimate the presence of crystalline molecules during nucleation events. On the other hand, the coarse-grained order parameter utilizing the 12th-order harmonic, Q_{12} , offers a clear and efficient method for identifying ice molecules (refer to Fig. R3). This issue may not significantly affect the estimation of nucleation rates in bulk systems. However, in cases where surfaces induce ordering with symmetries other than 6-fold, as in our scenario, using q_6 for biasing could potentially lead to incorrect conclusions.

We have incorporated the above information into the revised manuscript and have also referenced the paper kindly notified by the reviewer (Rosky et al. Chem Phys Lett 789, 139289 (2022)). These revisions are highlighted in blue. We have also included Fig. R3 as a part of Supplementary Fig. S1.

We believe that the above responses adequately address the reviewer's concerns and effectively convey the validity of our findings. We sincerely hope that the reviewer finds our revised manuscript suitable for publication in Nature Communications.

[1] Ghiringhelli, L. M., Valeriani, C., Meijer, E. j. & Frenkel, D. Phys. Rev. Lett. **99**, 055702 (2007). In this reference, the authors stated, “We used a local order parameter as introduced in Ref. [6]; rather than the 6th order harmonic, we found that a 3rd order harmonic can capture the correct symmetry for both graphite and diamond lattices.”

[2] Li, T., Donadio, D., Ghiringhelli, L. M. & Galli, G. Nat. Mater. **8**, 726-730 (2009). In this reference, the authors stated in the Methods section, “We use the local q_3 (ref. 24, the above [1]), a quantity that is highly sensitive to crystalline order, to identify Si crystalline clusters in the liquid.

[3] Russo, J., Romano, F. & Tanaka, H. Nat. Mater. **13**, 733-739 (2014).

Responses to the comments of Reviewer #3

I am satisfied with the author's response and, thus, recommend publication.

We are grateful to the reviewer for taking the time to review our revised manuscript again. We are delighted that the reviewer has recommended the publication of our paper in Nature Communications.

REVIEWERS' COMMENTS

Reviewer #1 (Remarks to the Author):

The authors have conducted additional brute-force simulations to confirm the validity of their conclusion. They also provided more analysis and reasoning to support their finding and try to reconcile the discrepancy between the current work and prior studies through q6 and Q12. Although this still isn't sufficient either to prove the relevance of their finding in a more relevant temperature or to explain the contradiction between the current and prior studies, I would say the current work has its own merit as it shows through unbiased simulation that film geometry could lead to a higher ice nucleation rate and it also points out the potential issue of q6. I find Fig. 3R to be interesting and I would suggest the authors to include this figure in the paper. Recent two-path model (JCP 158, 124501 (2023)) could help rationalize the discrepancy provided that it's proven that the pathway near the surface identified through q6 indeed leads to a lower or comparable rate than homogeneous nucleation.

Responses to the comments of Reviewer #1

The authors have conducted additional brute-force simulations to confirm the validity of their conclusion. They also provided more analysis and reasoning to support their finding and try to reconcile the discrepancy between the current work and prior studies through q6 and Q12. Although this still isn't sufficient either to prove the relevance of their finding in a more relevant temperature or to explain the contradiction between the current and prior studies, I would say the current work has its own merit as it shows through unbiased simulation that film geometry could lead to a higher ice nucleation rate and it also points out the potential issue of q6. I find Fig. 3R to be interesting and I would suggest the authors to include this figure in the paper. Recent two-path model (JCP 158, 124501 (2023)) could help rationalize the discrepancy provided that it's proven that the pathway near the surface identified through q6 indeed leads to a lower or comparable rate than homogeneous nucleation.

We first thank the reviewer for carefully reviewing our manuscript again. We are delighted to learn that the reviewer found our explanation reasonable. Following the reviewer's kind advice, we have cited the paper suggested by the reviewer. We have also included Fig. R3 in panels a-c in new Fig. 1.

We trust that our revised manuscript is now suitable for publication in Nature Communication.